# Conceptions of Consensual versus Non-Consensual Sexual Activity among Young People from Colombia

**DOI:** 10.3390/bs14100884

**Published:** 2024-10-01

**Authors:** Luis Enrique Prieto, Nieves Moyano

**Affiliations:** Department of Evolutionary and Developmental Psychology, University of Jaen, 23071 Jaen, Spain; lepp0002@red.ujaen.es

**Keywords:** sexual consent, qualitative approach, sexual communication, sexual agreement

## Abstract

Conceptions or ideas that couples hold about sexual consent could be a key factor in their communication, mutual respect, and the prevention of sexual violence. The multifaceted nature of sexual consent makes it a complex concept. The aim of the present study was to explore individuals’ ideas and understanding of sexual intercourse in two distinct contexts: consensual and non-consensual. We used a qualitative approach, adopting the methodology of thematic analysis. In total, 113 surveys obtained from the general population (76.1% women and 23.9% men aged 18 to 59 years) were studied. Two open-ended questions were asked about the general topic of sexual consent, where we distinguished sexual activity in which there is sexual consent vs. no consent. The phases of the thematic analysis approach were applied. For the consensual context, the following themes emerged: mutual reciprocity and respect; open, clear communication and agreements; and awareness and emotional well-being. For the non-consensual context, the following themes emerged: violence and sexual assault, absence and ambiguity of sexual consent, and lack of communication. All of these aspects should be considered in couples’ communication and sexual education to facilitate and improve sexual relationships and, in turn, prevent violence and sexual aggression.

## 1. Introduction

Sexual consent is one of the most important aspects of interpersonal relationships because, depending on how it is handled, it can affect couples’ relationships in a positive or negative way [1,2]. Permission to engage in a sexual relationship is influenced by several factors, such as the respect that partners feel for each other [2,3,4], the degree of communication [4,5,6], the expression of agreements and disagreements in the sexual relationship [7,8], education and sexual health [9,10,11], and the prevention of sexual violence and assault [12,13,14], among others.

What sexual consent means should be clear due to its implications for prevention and treatment of sexual assault [15,16]. Some of the definitions of sexual consent are reviewed in the following. Hickman and Muehlenhard [17] defined it as the free verbal or non-verbal communication of the willingness to participate in sexual activity. Lim and Roloff [18] emphasized that consent is a voluntary and conscious agreement to participate in sexual activity. Similarly, Beres [19] stated that sexual consent refers to a free willingness to engage in sexual relations with others, an aspect also presented in other definitions describing consent as a conscious desire to have sex with someone in a specific context [20]. Interestingly, other definitions highlight the dynamic role of consent by indicating that it is an ongoing and dynamic process that can encompass sexual encounters that go beyond simply “yes” or “no” [21]. In the last 5 years, consent has also included “desire” as part of its definition. For instance, Ref. [22] referred to it as the clear expression of a desire and strong willingness to have sex, shown through conversations or actions, while also avoiding any negative consequences. Finally, other authors commented that sexual consent refers to both an internal and external willingness to engage in sexual intercourse [23].

The timing of the establishment of sexual consent is also relevant. For example, sexual relationships are consensual activities where sexual consent is obtained through actions that occur between the couple at the beginning of the relationship [1]. Sexual consent is a process in which both partners are aware of the voluntary nature of each other’s participation in sexual activity [24]. Of the above definitions, only that of Glace et al. [24] conceived of consent as a process including three important aspects: the first is related to perceiving sexual consent as a continuous process in which monitoring the partner’s desire and feelings during sex is key, and the others are communicative sexuality and the existence of subtle coercion in sexual relations. As the authors indicated, our understanding of the concept of sexual consent has evolved, underscoring the relevance of considering it as a continuous process mediated by communicative sexuality and going beyond binary notions.

Sexual consent has been conceived of as a multidimensional construct in both quantitative (using self-reported measures) and qualitative research. From a quantitative approach, based on the most widely used scales, some authors [25] distinguished five dimensions, divided into two blocks. The first block consists of three attitudinal subscales that assess (a) perception of the need for consent, (b) beliefs about consent, and (c) experiences of communicating consent. The two behavioral subscales that constitute the second block evaluate (a) factors influencing consent and (b) attitudes towards consent. In contrast, sexual consent can be considered to encompass both internal and external consent [26]. Their position was adopted by several authors [4,10,23,27,28]. Internal consent is referred to as the personal desire to engage in a sexual relationship. This allows people to determine whether a sexual relationship is consensual or not. On the other hand, external consent refers to expressing to others the desire to have a sexual relationship with a partner; this manifests in both verbal and non-verbal ways [26,29]. Both the internal and external dimensions influence whether sexual relationships are considered satisfying and the prevention of sexual assault in an important way [2,28]. According to Glace et al. [24], sexual consent should be understood as a continuous process that leads to consent rather than just confirming that there is a signature on a consent form; in this respect, the clarity of information conveyed during sexual activity is important, as well as the voluntariness of participation, the ability of participants to ask questions and receive satisfactory answers, and perceptions of respect and consideration.

From a qualitative perspective, a recent review [30] indicated that future research should examine the variability in sexual consent among student samples, and that consolidating different research would lead to robust definitions of consent, which is a key aspect to move the research forward. Most of the qualitative studies exploring definitions of sexual consent described the following emerging themes: agreement, communication, mutual respect, safety, awareness, cultural and social influences, boundaries, and legal and political implications; see [31,32,33,34].

As detailed so far, multiple elements comprise the concept of sexual consent, and there are also several dimensions or factors that hinder people’s understanding of the development of respectful and consensual sexual competence, which may lead to erroneous or distorted attitudes to sexual relations [30]. Hence, it is important to explore conceptions of sexual consent among people in our country (Colombia), where communication norms and different interpretations may or may not promote an understanding of sexual consent [35]. Other areas that may be affected relate to legal challenges that people may face when they are not clear about what a consensual sexual relationship is [36].

Definitions of sexual consent are clearly influenced by sexual scripts [32], which suggests that there is a need to address socio-cultural norms/sexual scripts in the context of consent within the wider population [34]. In this regard, Colombia is a country in which the sexual double standard is clearly evident [37], reflected in the high prevalence of sexual violence. According to the [38], there were 3724 violent deaths among women in Colombia from various causes, with 27.28% corresponding to femicide; in other words, more than a quarter of violent deaths among women in Colombia were due to femicide.

In the context of sexual scripts, the research of [37] reveals that Colombian women show a tendency towards sexual shyness, thus endorsing a traditional dual sexual standard. Sexism embodies a biased disposition that, particularly with regard to Colombian women, forces them to asymmetrically construct the gender identities of women and men from the perspective of privilege in several dimensions of social behavior, including the realm of sexuality. Accordingly, in Colombia, women tend to prioritize female gender identity, predominantly shaped by social consensus, in order to delineate and internalize sexual behaviors that are considered permissible or impermissible for them [37]. This phenomenon is likely to exert a significant influence on crucial aspects such as sexual consent.

Considering the cultural differences in the definition of sexual consent and the lack of a clear consensus, the aim of the present study was to explore conceptions of sexual consent, based on definitions of consensual and non-consensual sexual activity, among Colombian men and women through a qualitative methodology.

## 2. Materials and Methods

### 2.1. Participants

Through non-probabilistic convenience sampling, 113 participants were selected, of whom 86 were women (76.1%) and 27 were men (23.9%). Their age ranged from 18 to 59 years (*M* = 27.62, *SD* = 12.21). The inclusion criteria were being Colombian and being at least 18 years old. An online survey was conducted. Convenience sampling is very common in qualitative studies [39,40], since it allows an exploration of the diverse meanings of concepts within a population [41]. Obtaining a large sample of participants increases the scope and sensitivity of the results [42]. Table 1 presents the socio-demographic variables evaluated in this study (educational level, marital status, and sexual orientation).

### 2.2. Measures

#### Sexual Consent Survey

A survey was designed, including a sociodemographic questionnaire and two open-ended questions: (1) For you, what is sexual activity in which there is sexual consent; and (2) For you, what is sexual activity in which there is no sexual consent? These questions focused on evaluating individuals’ conceptions of sexual consent. The questions were reviewed by three professionals trained in the area of sexuality and psychometric research, who provided their observations on the questions. Then a pilot study was administered to 15 individuals in order to check the understanding of the questions. No difficulties were found.

### 2.3. Procedure

The online survey was administered to the participants via a university platform, disseminated across several social networks. When the participants accessed the link, an informed consent document appeared, which included the following information: introduction to the research, objectives of the study, tasks to be performed, possible risks or disadvantages for the participant, benefits, the handling of personal data, and the possibility of withdrawing at any time. All participants had to accept the consent document to be able to access the questions. The anonymity and confidentiality of the information were guaranteed via pseudonymization and the elimination of identifying data. The study was approved by the ethical committee of Jaén University.

### 2.4. Data Analysis

The ATLAS.ti program (version 24.0.0) was used for the analysis. The six phases of the proposed process of thematic analysis [43] were applied: first, familiarization with the data/information; second, the generation of initial categories or codes; third, the search for themes; fourth, a review of the themes; fifth, defining and naming the themes; and sixth, production of the final report. Two members of the study team organized the codes and themes. The data were coded until no new codes appeared in the responses to the two questions; that is, until information saturation had been achieved [44].

## 3. Results

Iterative qualitative coding of the interviews was used to generate the research results. A codebook was created to organize the data, and it was subsequently reviewed by two members of the study team. The coding team consisted of two coders who reviewed the results, coding, and discrepancies among the 113 surveys. Thematic analysis [43] was applied to the coded data, and themes and subthemes were obtained. Thematic analysis is an adaptative method of qualitative data analysis used to identify, organize, describe, and report content based on a summary of the findings [43].

According to the definition of saturation in qualitative analyses in [44], our surveys continued until the research team determined that the study data had reached theoretical saturation. Saturation was considered to have been reached when the researchers believed that similar data on a topic were being found among the participants such that there was empirical certainty that the category was saturated. The surveys were conducted over a period of 3 months, and the sampling, data collection, and analysis procedures were conducted simultaneously over a period of 4 months until saturation of the data related to these questions was reached.

The resulting codebook was reviewed by two members of the research team and finalized to include, in relation to Question 1, three main codes pertaining to consent: “mutual reciprocity and respect”, “open, clear communication and agreements”, and “awareness and emotional well-being”. For Question 2, three main codes appeared: “violence and sexual assault”, “absence or ambiguity of sexual consent”, and “lack of communication”.

The coding team, consisting of two coders, met regularly to discuss the findings of the coding and to address discrepancies. The percentage of agreement and [45] kappa coefficient were calculated to assess inter-rater reliability across all of the participant responses included. For consent codes related to the definition of consent, conveyance of consent, and continuous consent, the inter-rater reliability was very good (Cohen’s κ = 0.72–0.85) with a percentage of agreement of 0.82%. The codes were reviewed and summarized according to commonly described attitudes and shared experiences among the participants. Thematic analysis methods [43] were applied to the coded data to derive the main themes and subthemes. Thematic analysis is an adaptable method of qualitative data analysis used to identify, organize, describe, and report themes based on summarized findings [43]. Unlike when theoretical frameworks are used, thematic analysis allows the main findings or themes to emerge directly from trends in the data [43]. Despite the differences in the sample sizes of men and women, it can be observed that the categories or themes obtained from the thematic analysis were the same for each sample and for the total number of participants, which was ratified with quotations from each sex for each theme.

Initially, we present the information obtained in the coded document tables, then we explain the codes generated according to the information obtained and some quotes that support them, followed by the generated networks and an explanation of the themes constructed from the codes. All the information was pseudonymized.

### 3.1. What Is Sexual Activity in Which There Is Sexual Consent?

Regarding Question 1 (“For you, what is sexual activity in which there is sexual consent?”), we started from the six codes listed in Table 2. According to the analysis, definitions, and relationships of the codes, the decision was made to group them into three emergent general themes or categories as follows.

Theme 1: Mutual reciprocity and respect was defined as the free approval of people to engage in sexual activity, such that boundaries are respected, as a continuous process where the importance of respect for the other is highlighted. This theme included two subthemes: mutual consent and reciprocity, and respect.

Subtheme 1 was the most frequently commented-on theme (84 citations). It refers to mutual and reciprocal approval given by people to engage in sexual activity, such that boundaries are respected. Most definitions indicated that this should be an ongoing process.


*“Both people in the couple ask about and accept sexual boundaries, both individually and as a couple, thus respecting their relationship.”*
(Alexa, 19)


*“It is a relationship where both parties accept and agree to perform certain sexual acts, and there is total freedom to participate in these.”*
(Manuela, 19)

Respect was defined as consideration, deference, and empathy for the partner in the sexual relationship.


*“It is a relationship in which there is trust, respect, and integrity.”*
(Silvia, 19)


*“Where there is respect in the relationship and consent on the part of both parties.”*
(Luisa, 21)

Theme 2: Open, clear communication and agreements in sexual relations were defined as open and clear verbal and non-verbal agreements or pacts pertaining to sexual relations where boundaries and expectations are established. This theme included subthemes of communication and agreement.

Communication refers to the two parties involved openly and honestly expressing what is and is not allowed in the sexual relationship. Some quotes on this subject follow.


*“A relationship where there is consent, where the people involved talk about what they like, what they are willing or unwilling to do, and what they don’t want, and these agreements clearly apply.”*
(Laura, 21)


*“A relationship where there is communication, especially in the sexual sphere, is very important and very valuable, since they can communicate about what is accepted and what is not for both individuals; in this way, they will not have emotional or physical problems, and they will enjoy the relationship more.”*
(Daniel, 19)

Regarding agreement on relations, the codes generated from the information can be described as follows. Agreement refers to an agreement, pact, or consultation that people have about the sexual acts to be performed, in which there are certain rules. Some quotes from our participants that support this concept follow.


*“Sexual consent is a written or verbal pact where certain characteristics of the sexual and affective relationship that two people can have defined, and rules are established depending on whether there is exclusivity or not.”*
(Eduardo, 19)


*“It is a relationship where you have, from the beginning, clear rules of what can or cannot occur during a sexual act, besides always being in continuous communication with the partner on the subject.”*
(Carla, 28)

Theme 3: Awareness and emotional well-being. This category can be defined as the importance of being aware of the actions being carried out and enjoying sexual relations in a healthy way, promoting the well-being of both parties. This theme included the subthemes of awareness and comfort during the sexual event.

*Awareness of the act.* This can be understood as the understanding that people have about what it means to have sexual relations, where there must be approval and agreement by both parties:


*“Where both parties agree in a conscious manner, i.e., without substances such as alcohol and drugs, and where both parties are in physical and psychological agreement.”*
(Gloria, 19)


*“A relationship between two people old enough to openly and consciously express their agreement to such a sexual relationship.”*
(Cecilia, 19)

Comfort during the sexual event referred to the tastes, interests, or pleasure of both parties engaging in a sexual relationship, evidenced by reports such as the following.


*“It is a relationship where the two parties feel comfortable and voluntarily agree to maintain a sexual relationship, which must be terminated if either party feels uncomfortable or wants to withdraw.”*
(Marcos, 23)

*“A relationship in which the members of the couple explicitly accept and give their consent for the development of a sexual encounter that, by consensus, they have considered appropriate and with which they feel comfortable.”* (Rocio, 43)

### 3.2. What Is Sexual Activity in Which There Is No Sexual Consent?

As shown in Table 3, three main themes, divided into subthemes, were generated from the participants’ responses. The most frequent code was the subtheme of so-called lack of mutual consent, with a total of 50 citations, followed by violence, with 48 citations, lack of communication (16 citations), and finally disrespect for boundaries, pressure and sexual coercion, state of vulnerability, and doubt about consent, which had lower frequencies.

Theme 1: Violence and sexual assault. This category included codes for violence, disrespect for boundaries, and pressure and sexual coercion. The subtopic of violence refers to more extreme cases of lack of consent, where attempted rape, rape, or physical and psychological aggression may be directed towards a person who refuses to have sexual relations.


*“Simply put, a relationship where there is no sexual consent is rape, since no input or affirmation for you to engage has been given.”*
(María, 19)


*“Where one of the persons imposes themselves or forces the other to perform an act by verbally and physically assaulting him/her.”*
(Roberta, 19)


*“This is where a person is forced to engage in, or is uncomfortable with, certain actions but is not able to express this or it is simply not taken into account. This can also apply when someone exerts so much pressure that the only way to ‘get out of it’ is to give in to the person regardless of having previously told them no on multiple occasions.”*
(Efigenia, 19)


*“Where some limit is exceeded or broken in order to please only one person; I feel this is some kind of manipulation or obligation.”*
(María, 26)

Theme 2: Absence or ambiguity of sexual consent, which refers to a lack of permission from one of the members of the couple to have sexual relations; this includes the codes lack of mutual consent, state of vulnerability, and doubt about sexual consent. The first subtheme pertains to the violation of previously established agreements (boundaries) regarding what one is willing to give and what is non-negotiable when performing a sexual act. Some citations that represent this theme follow.


*“A non-consensual sexual relationship is one in which a couple’s pre-established boundaries are violated.”*
(Alejandra, 19)


*“It is a sexual interaction in which the persons involved are not fully aware of, do not accept, or transgress the boundaries and scope of the interaction at the physical, verbal, emotional, or symbolic levels.”*
(Jorge, 40)


*“An abusive relationship where one of the parties is forced into sexual acts that she does not want to perform.”*
(Graciela, 19)


*“Where one of the two people does not want to have sexual relations.”*
(Alejandra, 19)

The following code pertains to a type of vulnerability where people cannot make decisions about whether or not to have sexual relations due to a state of altered consciousness, such as being under the effects of psychoactive substances or acute emotional distress.


*“When ‘NO’ is not respected, and where the other person is abused when he/she has taken substances such as alcohol or drugs.”*
(Diana, 19)


*“On the other hand, in a relationship without sexual consent, one of the persons is not aware and cannot confirm whether or not he/she wants to perform the act, or one of the persons forces the other to perform it.”*
(Fanny, 18).

Doubt about consent refers to the insecurity that a person may have when having sexual relations, possibly related to the fact that there was no real or clear consent to engage in a sexual relationship.


*“Where you have a sexual relationship without being sure.”*
(Manuel, 54)


*“When one of the parties involved in the relationship is not in full agreement on a sexual practice.”*
(Armando, 57)

Theme 3: Lack of communication refers to open and clear verbal and non-verbal agreements or pacts on sexual relations, where boundaries and expectations are established. A subtheme refers to a lack of communication in the context of sexual relations, where there are no agreements or boundaries in the sexual relationship.


*“Where there is no communication in the relationship or in the sexual sphere, there will be problems, such as physical, mental, and emotional problems, among others, where both people are affected albeit one more than the other.”*
(Carlos, 19)


*“It is a relationship in which no questions are asked about each other’s sexual well-being, nor is there any communication about it.”*
(Evelyn, 19)

## 4. Discussion

This study aimed to explore Colombian nationals’ conceptions of sexual consent, using a qualitative methodology with participants aged 18 and above. Two open-ended questions were used to explore opinions of consensual and non-consensual sexual relationships. Using thematic analysis, three themes were identified for each question, resulting in six themes in total. These themes reflected the participants’ beliefs about the concept of sexual consent.

Regarding the characterization of consensual context, three themes emerged: mutual reciprocity and respect; open, clear communication and agreements; and awareness and emotional well-being. The notion that the first theme, mutual reciprocity and respect, is the key factor in sexual consent has been supported by several studies [46,47]. Therefore, our findings corroborated the importance of mutual consent in the sexual relationship and also suggested that we must go further by considering respect as a more important variable in sex education than is the case currently. Other authors included mutuality as an ethical norm in sexual relationships but commented that it may not be sustainable, especially in sex education in schools; we should work towards addressing this situation in schools [48]. Other variables in addition to mutuality were proposed by Lab et al. [46], who commented that aspects such as care and loving attention between the members of the couple should be recognized; this is fundamental for the prevention of sexual violence and mutual consent. Within this theme, the subtheme of respect was in line with the previous findings of [49], according to whom, respect is closely related to sexual consent. It is important to promote an appropriate culture of sexual consent for healthy sexual relationships. Respect is even more fundamental than sexual consent and other variables associated with sexual relationships, i.e., the culture of respect subsumes these two aspects [47]. In this sense, respect in sexual relations promotes the well-being of the couple and autonomy in decision-making regarding these activities [50]. This highlights the association between autonomy and sexual consent, which allows the couple to assert their rights without coercion, and allows us to understand the effects of power, desire, and social norms in the conception of sexual consent [51].

The second theme to emerge in this study, open, clear communication and agreement, can be subdivided into two subthemes: communication and agreements in relationships. Couples’ communication is among the variables that we consider to be most relevant to sexual consent, as corroborated by several studies. Effective education on sexual communication within communities is fundamental to generate a culture of respect and informed consent [52]. Improving the openness and clarity of communication regarding sexual consent will certainly lead to better understanding and respect in sexual interactions and positive sexual experiences for couples [53,54]. Another aspect relevant to sexual consent found in this study is the so-called agreement in sexual relations, where agreements reached by couples facilitate or prevent sexual relations. Discrepancies in agreements are related to greater risk in the context of sexual behavior [7]. Crucially, relationship agreements can influence sexual consent, as they can promote understanding and communication of the sexual boundaries that couples are willing to operate within [55].

The third theme is called awareness and emotional well-being. This includes the awareness that individuals have when performing a sexual act and an assessment of the level of comfort during the act. It is essential that young people have sexual awareness because it facilitates informed decision-making, allowing them to avoid pregnancies or risks in their sex lives; that is, it has a strong effect on sexual consent [56]. Classroom interventions should focus on improving students’ understanding of sexual awareness, as was achieved with the Media Aware program applied in the United States with the aim of improving critical thinking in relation to media messages and skills related to sexual health [57].

Three themes also emerged regarding non-consensual context: violence and sexual assault, absence or ambiguity of sexual consent, and lack of communication. Sexual violence and assault are among the most profound consequences of the absence or misunderstanding of sexual consent. In the study of [58], carried out in Norway, the following determinants of sexual violence (among others) were found: sexual risk behavior, alcohol intoxication, pornography, rape stereotypes, and previous victimization. Some of these agreed with the present study, and there are other variables that merit analysis in subsequent studies. Factors that increase the possibility of sexual assault include gender, alcohol consumption, sexual disorders, environmental influences, and religious beliefs [59].

The second theme of Question 2 was the absence or ambiguity of sexual consent. We found that psychological and situational vulnerability can be associated with a lack of sexual consent, in agreement with the results of the study of [60], which similarly proposed a relationship between psychological vulnerability and a lack of sexual consent. Women victims of sexual assault present ideas and attitudes that move away from control towards maintaining sexual consent [60]. Some of the psychological states that can lead to sexual aggression are coercion and threats, which can occur with varying degrees of severity in the relationship, as stated by [61].

The third theme of the second question is called lack of communication. A study carried out on Danish youths found that sexual consent had three interpretive components: consent as an agreement between individuals, consent as a normative practice between different groups, and the influence of alcohol on the lack of consent [33]. These results agreed with the findings of the present study of Colombian youths, which proposed that states of vulnerability and communication are influential factors in sexual consent. In a qualitative study, sexual scripts were found to be important in sexual consent: men perceived resistance from women as symbolic, and they adhered better to the sexual chastity norm of the community [62]. Assertiveness and communication are important aspects of relationships, which accord with one of the subthemes emerging in this study, namely communication. This aspect is particularly relevant when analyzing the predictors of sexual consent.

Our findings are in line with previous qualitative research [31], in which focus groups were used. They identified six relevant themes regarding the attitudes and behaviors of students towards sexual consent: communication, mutual respect, awareness and education, cultural and social influences, personal boundaries and comfort levels, and legal and political implications. These findings correspond with those of the present research, highlighting the importance of communication [63,64,65], respect for the partner, awareness and emotional well-being, and personal boundaries, as well as aspects that we consider important but which did not emerge in our analyses, such as cultural influences and legal implications. When comparing our study with that of [34] on sexual consent in Canadian students, an adequate understanding of sexual scripts related to legal aspects was found, but the relationship was not as clear in various circumstances of ambiguous sexual consent. These aspects appeared to be important in the context of our study, which did not consider the legal aspect of consent.

The present research has some limitations. The sample comprised a relatively young and predominantly heterosexual population, in which more women than men participated. This should be considered before generalizing the results. Future research should include a population with a wider age range, and various gender identities or sexual orientations. Another issue to consider is the way in which the data were coded. Although the subthemes and themes emerged from individual interpretations and joint coding, a degree of subjectivity in the construction of the codes may have been possible.

In conclusion, this study of conceptions of sexual consent among the Colombian population has revealed major factors that are not only important in this specific context, but also have broader implications that transcend borders and cultures, as recently pointed out [30]. These factors underscore the complexity of sexual consent and its ties to psychological, social, and cultural influences. Therefore, these findings should be considered in various contexts, including sexual relationships in couples, promoting open communication and mutual respect; sexual education in schools, focusing on the relevance of the relationship between sexual consent and autonomy; public policies, i.e., designing effective strategies to prevent violence and sexual assault; and the prevention of violence and sexual assault in general, recognizing the importance of sexual education for the generation of a culture of respect and non-violence.

## Figures and Tables

**Table 1 behavsci-14-00884-t001:** Description of socio-demographic variables.

	*n*	%		*n*	%
Marital status			Sexual orientation		
Married	19	16.81	Bisexual	17	15.04
Divorced or separated	5	4.42	Exclusively heterosexual	84	74.33
Single	80	70.79	Exclusively homosexual	2	1.77
Free union	9	7.96	Pansexual	1	0.88
Education			Predominantly heterosexual with homosexual contacts	5	4.42
Secondary education	17	15.04	Predominantly homosexual with heterosexual contacts	1	0.88
Technical/technological training	2	1.77	Predominantly homosexual with more than sporadic contacts	1	0.88
Postgraduate education	27	23.89	Missing data	2	1.77
University	67	59.29			

**Table 2 behavsci-14-00884-t002:** Themes, subthemes, and frequencies for the question “For you, what is a sexual activity in which there is sexual consent?”.

	Frequency
Mutual reciprocity and respect	
Mutual consent and reciprocity	84
Respect	15
Open, clear communication and agreement	
Communication	27
Agreements on relations	23
Awareness and emotional well-being	
Awareness	37
Comfort during sexual event	10

**Table 3 behavsci-14-00884-t003:** Themes, subthemes, and frequencies for the question “For you, what is a sexual activity in which there is no sexual consent?”.

	Frequency
Violence and sexual assault	
Violence	48
Disrespect for boundaries	14
Pressure and sexual coercion	14
Absence or ambiguity of sexual consent	
Lack of mutual consent	50
State of vulnerability	9
Doubt about consent	6
Lack of communication	
Communication	16

## Data Availability

Data are contained within the article.

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
