# Peer review of "Conceptions of Consensual versus Non-Consensual Sexual Activity among Young People from Colombia"

_behavsci, 2024, doi:10.3390/bs14100884_

Round 1

Reviewer 1 Report

Comments and Suggestions for Authors

The study explores how individuals perceive sexual consent in both consensual and non-consensual contexts, using thematic analysis of 113 survey responses (76.1% women and 23.9% men, aged 18 to 59). The study addresses an important social issue: sexual consent. The inclusion of themes related to emotional well-being and communication highlights the multidimensionality of consent, making the study highly relevant to educational efforts and violence prevention strategies. For consensual situations, themes such as mutual respect, clear communication, and emotional well-being emerged. In non-consensual situations, themes included violence, ambiguity of consent, and lack of communication. The findings emphasize the importance of addressing these themes in couple communication and sexual education to improve relationships and prevent sexual violence. In the abstract, the study's purpose is well defined: to explore conceptions of sexual consent in consensual versus non-consensual contexts. This focus on the differences between the two contexts is crucial, and it is well-articulated. The qualitative approach using thematic analysis is appropriate for exploring the nuanced perceptions of individuals. The survey sample is relatively large (113 participants), which adds credibility, but the demographic details (mostly women) should be considered when interpreting results.

The introduction provides a comprehensive background on the concept of sexual consent, framing it as a crucial element in interpersonal relationships. It reviews various definitions of sexual consent from different scholars, highlighting that consent is a dynamic, ongoing process beyond simple "yes" or "no" responses. It also touches on internal and external consent and their significance in preventing sexual violence. It  effectively emphasizes the multidimensional nature of consent, supported by both quantitative and qualitative research. Finally, it contextualizes the study within the Colombian cultural framework, where norms and sexual scripts, along with a high prevalence of gender violence, may complicate clear understandings of consent.

The introduction could benefit from a stronger focus on how cultural differences specifically shape consent perceptions in Colombia. The references to various studies add credibility, but some details are redundant and could be streamlined.

Methods and Materials. This section provides a clear and structured overview of the study's materials and methods, but a few points are worth noting:

·        The use of non-probabilistic convenience sampling is appropriate for a qualitative study, as it allows for the exploration of diverse viewpoints. However, the heavy gender imbalance (76.1% women) may influence the findings, which could have been addressed by either acknowledging this as a limitation or attempting to obtain a more gender-balanced sample.

·        The socio-demographic breakdown of participants (age, marital status, sexual orientation, education) provides useful context for interpreting the results, and the table aids in clarity. However, some categories like "predominantly homosexual with more than sporadic contacts" could benefit from clearer definitions, as these distinctions may not be universally understood.

·        The two open-ended questions are well designed to capture individuals' perceptions of sexual consent. However, additional context on how these questions were validated or tested before administration would strengthen the study’s reliability.

The Results section offers a clear and structured presentation of findings derived from qualitative analysis. The iterative process of coding, with two coders reviewing and resolving discrepancies, ensures rigor and reliability. The application of thematic analysis aligns with the study's qualitative nature and allows for a rich exploration of participant experiences and perceptions. The emergent themes for both questions are well-categorized and clearly supported by participant quotes. For Question 1 (sexual activity with consent), the key themes of "Mutual Reciprocity and Respect," "Open, Clear Communication and Agreements," and "Awareness and Emotional Well-being" provide a nuanced understanding of consent. For Question 2 (absence of consent), themes such as "Violence and Sexual Assault" and "Absence or Ambiguity of Consent" highlight the participants' perspectives on violations of consent, which are critical for exploring negative experiences.

The discussion section of this study effectively ties together the key findings related to conceptions of sexual consent among Colombian nationals. It highlights three primary themes for consensual contexts—mutual reciprocity and respect, open communication, and emotional awareness—and emphasizes the importance of respect as a foundational element in sexual relationships, aligning with existing literature. For non-consensual contexts, themes like violence, ambiguity of consent, and lack of communication are well explored, providing a clear connection to situational and psychological vulnerabilities. The study also compares its findings with other research, showing both similarities and gaps, such as the absence of legal and cultural dimensions in the current analysis.

The limitations are noted, particularly the homogeneity of the sample, but the authors suggest directions for future research, advocating for more diverse populations. The conclusion successfully underscores the practical implications, stressing the need for improved sex education, policy interventions, and a culture of respect.

Recommendations

The article presents a thorough exploration of sexual consent conceptions among Colombian nationals, using qualitative methodology to analyze participants' responses. The thematic analysis identifies six key themes: mutual reciprocity and respect, communication, emotional well-being, violence, ambiguity of consent, and lack of communication. The findings align with existing literature, while also contributing new insights, particularly regarding the centrality of respect in sexual consent.

The study is well-organized, and the discussion connects the results to broader psychological, social, and cultural variables, offering valuable implications for sexual education, relationship dynamics, and public policy. The authors acknowledge limitations, such as the sample's lack of diversity, and suggest areas for future research.

While the article has a strong foundation and contributes meaningfully to the discourse on sexual consent, it would benefit from addressing legal and cultural dimensions more explicitly, which were noted as absent. However, the article’s findings are relevant and impactful, making it a valuable addition to the literature. With minor revisions, particularly around the study's limitations, the article is worth publishing.

Comments on the Quality of English Language

Overall Evaluation:

The English is generally understandable, but some sentences are overly complex and could be simplified to enhance clarity and flow. There are a few minor issues with subject-verb agreement and awkward phrasing. Addressing these would improve the text's grammatical accuracy. The tone is mostly appropriate for an academic article, but certain parts can benefit from increased precision and conciseness. The ideas are well-structured, but occasionally, the sentence structure could be more direct and less redundant.

With some editing, particularly focusing on conciseness and clarity, the English in the article would be of a high standard.

Objective and Methodology (Lines 306-312):

  • The sentence "The objective of this research was to explore the conceptions of sexual consent among Colombian nationals aged at least 18 years through a qualitative methodology." could be shortened for simplicity:
    Suggestion: "This study aimed to explore Colombian nationals' conceptions of sexual consent, using a qualitative methodology with participants aged 18 and above."
  • The phrase "The participants' responses were analyzed using the thematic analysis technique, with three themes being identified for each question, for a total of six themes." could be clearer. Suggestion: "Using thematic analysis, three themes were identified for each question, resulting in six total themes."

Conclusion (Lines 409-419):

  • The sentence "These factors highlight the complexity of sexual consent and its interconnection with various psychological, social, and cultural variables" could be more precise. Suggestion: "These factors underscore the complexity of sexual consent and its ties to psychological, social, and cultural influences."
  • The phrase "For this reason, it is important to consider these findings in various contexts..." can be shortened. Suggestion: "Therefore, these findings should be considered in various contexts, including..."

Author Response

The study explores how individuals perceive sexual consent in both consensual and non-consensual contexts, using thematic analysis of 113 survey responses (76.1% women and 23.9% men, aged 18 to 59). The study addresses an important social issue: sexual consent. The inclusion of themes related to emotional well-being and communication highlights the multidimensionality of consent, making the study highly relevant to educational efforts and violence prevention strategies. For consensual situations, themes such as mutual respect, clear communication, and emotional well-being emerged. In non-consensual situations, themes included violence, ambiguity of consent, and lack of communication. The findings emphasize the importance of addressing these themes in couple communication and sexual education to improve relationships and prevent sexual violence. In the abstract, the study's purpose is well defined: to explore conceptions of sexual consent in consensual versus non-consensual contexts. This focus on the differences between the two contexts is crucial, and it is well-articulated. The qualitative approach using thematic analysis is appropriate for exploring the nuanced perceptions of individuals. The survey sample is relatively large (113 participants), which adds credibility, but the demographic details (mostly women) should be considered when interpreting results.

The introduction provides a comprehensive background on the concept of sexual consent, framing it as a crucial element in interpersonal relationships. It reviews various definitions of sexual consent from different scholars, highlighting that consent is a dynamic, ongoing process beyond simple "yes" or "no" responses. It also touches on internal and external consent and their significance in preventing sexual violence. It effectively emphasizes the multidimensional nature of consent, supported by both quantitative and qualitative research. Finally, it contextualizes the study within the Colombian cultural framework, where norms and sexual scripts, along with a high prevalence of gender violence, may complicate clear understandings of consent.

The introduction

in the Introduction is marked with yellow color

Methods and Materials. This section provides a clear and structured overview of the study's materials and methods, but a few points are worth noting:

  • The use of non-probabilistic convenience samplingis appropriate for a qualitative study, as it allows for the exploration of diverse viewpoints. However, the heavy gender imbalance (76.1% women) may influence the findings, which could have been addressed by either acknowledging this as a limitation or attempting to obtain a more gender-balanced sample.

  • The socio-demographic breakdown of participants (age, marital status, sexual orientation, education) provides useful context for interpreting the results, and the table aids in clarity. However, some categories like "predominantly homosexual with more than sporadic contacts" could benefit from clearer definitions, as these distinctions may not be universally understood.

Response: These categories were based on the Kinsey scale. The Kinsey scale, conceptualized by Alfred Kinsey, serves as an instrument to assess an individual's sexual orientation along a continuum ranging from fully heterosexual to fully homosexual. This scale recognizes the existence of sexual orientation as a spectrum, rather than adhering to a dichotomous categorization.

  • The two open-ended questionsare well designed to capture individuals' perceptions of sexual consent. However, additional context on how these questions were validated or tested before administration would strengthen the study’s reliability.

Response: a paragraph was added in the section on instruments where the validation by judges and the administration to 5 people to test the understanding of the questions is commented.

The Results section offers a clear and structured presentation of findings derived from qualitative analysis. The iterative process of coding, with two coders reviewing and resolving discrepancies, ensures rigor and reliability. The application of thematic analysis aligns with the study's qualitative nature and allows for a rich exploration of participant experiences and perceptions. The emergent themes for both questions are well-categorized and clearly supported by participant quotes. For Question 1 (sexual activity with consent), the key themes of "Mutual Reciprocity and Respect," "Open, Clear Communication and Agreements," and "Awareness and Emotional Well-being" provide a nuanced understanding of consent. For Question 2 (absence of consent), themes such as "Violence and Sexual Assault" and "Absence or Ambiguity of Consent" highlight the participants' perspectives on violations of consent, which are critical for exploring negative experiences.

The discussion section of this study effectively ties together the key findings related to conceptions of sexual consent among Colombian nationals. It highlights three primary themes for consensual contexts—mutual reciprocity and respect, open communication, and emotional awareness—and emphasizes the importance of respect as a foundational element in sexual relationships, aligning with existing literature. For non-consensual contexts, themes like violence, ambiguity of consent, and lack of communication are well explored, providing a clear connection to situational and psychological vulnerabilities. The study also compares its findings with other research, showing both similarities and gaps, such as the absence of legal and cultural dimensions in the current analysis.

The limitations are noted, particularly the homogeneity of the sample, but the authors suggest directions for future research, advocating for more diverse populations. The conclusion successfully underscores the practical implications, stressing the need for improved sex education, policy interventions, and a culture of respect.

Recommendations

The article presents a thorough exploration of sexual consent conceptions among Colombian nationals, using qualitative methodology to analyze participants' responses. The thematic analysis identifies six key themes: mutual reciprocity and respect, communication, emotional well-being, violence, ambiguity of consent, and lack of communication. The findings align with existing literature, while also contributing new insights, particularly regarding the centrality of respect in sexual consent.

The study is well-organized, and the discussion connects the results to broader psychological, social, and cultural variables, offering valuable implications for sexual education, relationship dynamics, and public policy. The authors acknowledge limitations, such as the sample's lack of diversity, and suggest areas for future research.

While the article has a strong foundation and contributes meaningfully to the discourse on sexual consent, it would benefit from addressing legal and cultural dimensions more explicitly, which were noted as absent. However, the article’s findings are relevant and impactful, making it a valuable addition to the literature. With minor revisions, particularly around the study's limitations, the article is worth publishing.

in the Introduction is marked with yellow color

Comments on the Quality of English Language

Overall Evaluation:

The English is generally understandable, but some sentences are overly complex and could be simplified to enhance clarity and flow. There are a few minor issues with subject-verb agreement and awkward phrasing. Addressing these would improve the text's grammatical accuracy. The tone is mostly appropriate for an academic article, but certain parts can benefit from increased precision and conciseness. The ideas are well-structured, but occasionally, the sentence structure could be more direct and less redundant.

With some editing, particularly focusing on conciseness and clarity, the English in the article would be of a high standard.

Objective and Methodology (Lines 306-312):

  • The sentence "The objective of this research was to explore the conceptions of sexual consent among Colombian nationals aged at least 18 years through a qualitative methodology." could be shortened for simplicity:
    Suggestion: "This study aimed to explore Colombian nationals' conceptions of sexual consent, using a qualitative methodology with participants aged 18 and above."
  • themes were identified for each question, resulting in six total themes."

Response: the suggestions were accepted and the changes were made to the text of the article.

Conclusion (Lines 409-419):

  • The sentence "These factors highlight the complexity of sexual consent and its interconnection with various psychological, social, and cultural variables"
  • should be considered in various contexts, including..."

Response: the suggestions were accepted and the changes were made to the text of the article.

REVIEWER 2

Enjoyed reading this well written article. In the discussion - the highlight of how respect is even more fundamental then consent is a great point 

Comments on the Quality of English Language

The minor grammatical/spelling errors are listed below:

line 41 - dYnamic

Response: the suggestions were accepted and the changes were made to the text of the article.

Response: a paragraph was added in the section on instruments where the validation by judges and the administration to 5 people to test the understanding of the questions is commented.

Response: the suggestions were accepted and the changes were made to the text of the article.

Response: the suggestions were accepted and the changes were made to the text of the article.

Reviewer 2 Report

Comments and Suggestions for Authors

Enjoyed reading this well written article. In the discussion - the highlight of how respect is even more fundamental then consent is a great point 

Comments on the Quality of English Language

The minor grammatical/spelling errors are listed below:

line 41 - dYnamic

line 50 actions that occur BETWEEN couple

line 386 which accord- not accordS

Author Response

The study explores how individuals perceive sexual consent in both consensual and non-consensual contexts, using thematic analysis of 113 survey responses (76.1% women and 23.9% men, aged 18 to 59). The study addresses an important social issue: sexual consent. The inclusion of themes related to emotional well-being and communication highlights the multidimensionality of consent, making the study highly relevant to educational efforts and violence prevention strategies. For consensual situations, themes such as mutual respect, clear communication, and emotional well-being emerged. In non-consensual situations, themes included violence, ambiguity of consent, and lack of communication. The findings emphasize the importance of addressing these themes in couple communication and sexual education to improve relationships and prevent sexual violence. In the abstract, the study's purpose is well defined: to explore conceptions of sexual consent in consensual versus non-consensual contexts. This focus on the differences between the two contexts is crucial, and it is well-articulated. The qualitative approach using thematic analysis is appropriate for exploring the nuanced perceptions of individuals. The survey sample is relatively large (113 participants), which adds credibility, but the demographic details (mostly women) should be considered when interpreting results.

The introduction provides a comprehensive background on the concept of sexual consent, framing it as a crucial element in interpersonal relationships. It reviews various definitions of sexual consent from different scholars, highlighting that consent is a dynamic, ongoing process beyond simple "yes" or "no" responses. It also touches on internal and external consent and their significance in preventing sexual violence. It effectively emphasizes the multidimensional nature of consent, supported by both quantitative and qualitative research. Finally, it contextualizes the study within the Colombian cultural framework, where norms and sexual scripts, along with a high prevalence of gender violence, may complicate clear understandings of consent.

The introduction

in the Introduction is marked with yellow color

Methods and Materials. This section provides a clear and structured overview of the study's materials and methods, but a few points are worth noting:

  • The use of non-probabilistic convenience samplingis appropriate for a qualitative study, as it allows for the exploration of diverse viewpoints. However, the heavy gender imbalance (76.1% women) may influence the findings, which could have been addressed by either acknowledging this as a limitation or attempting to obtain a more gender-balanced sample.

  • The socio-demographic breakdown of participants (age, marital status, sexual orientation, education) provides useful context for interpreting the results, and the table aids in clarity. However, some categories like "predominantly homosexual with more than sporadic contacts" could benefit from clearer definitions, as these distinctions may not be universally understood.

Response: These categories were based on the Kinsey scale. The Kinsey scale, conceptualized by Alfred Kinsey, serves as an instrument to assess an individual's sexual orientation along a continuum ranging from fully heterosexual to fully homosexual. This scale recognizes the existence of sexual orientation as a spectrum, rather than adhering to a dichotomous categorization.

  • The two open-ended questionsare well designed to capture individuals' perceptions of sexual consent. However, additional context on how these questions were validated or tested before administration would strengthen the study’s reliability.

Response: a paragraph was added in the section on instruments where the validation by judges and the administration to 5 people to test the understanding of the questions is commented.

The Results section offers a clear and structured presentation of findings derived from qualitative analysis. The iterative process of coding, with two coders reviewing and resolving discrepancies, ensures rigor and reliability. The application of thematic analysis aligns with the study's qualitative nature and allows for a rich exploration of participant experiences and perceptions. The emergent themes for both questions are well-categorized and clearly supported by participant quotes. For Question 1 (sexual activity with consent), the key themes of "Mutual Reciprocity and Respect," "Open, Clear Communication and Agreements," and "Awareness and Emotional Well-being" provide a nuanced understanding of consent. For Question 2 (absence of consent), themes such as "Violence and Sexual Assault" and "Absence or Ambiguity of Consent" highlight the participants' perspectives on violations of consent, which are critical for exploring negative experiences.

The discussion section of this study effectively ties together the key findings related to conceptions of sexual consent among Colombian nationals. It highlights three primary themes for consensual contexts—mutual reciprocity and respect, open communication, and emotional awareness—and emphasizes the importance of respect as a foundational element in sexual relationships, aligning with existing literature. For non-consensual contexts, themes like violence, ambiguity of consent, and lack of communication are well explored, providing a clear connection to situational and psychological vulnerabilities. The study also compares its findings with other research, showing both similarities and gaps, such as the absence of legal and cultural dimensions in the current analysis.

The limitations are noted, particularly the homogeneity of the sample, but the authors suggest directions for future research, advocating for more diverse populations. The conclusion successfully underscores the practical implications, stressing the need for improved sex education, policy interventions, and a culture of respect.

Recommendations

The article presents a thorough exploration of sexual consent conceptions among Colombian nationals, using qualitative methodology to analyze participants' responses. The thematic analysis identifies six key themes: mutual reciprocity and respect, communication, emotional well-being, violence, ambiguity of consent, and lack of communication. The findings align with existing literature, while also contributing new insights, particularly regarding the centrality of respect in sexual consent.

The study is well-organized, and the discussion connects the results to broader psychological, social, and cultural variables, offering valuable implications for sexual education, relationship dynamics, and public policy. The authors acknowledge limitations, such as the sample's lack of diversity, and suggest areas for future research.

While the article has a strong foundation and contributes meaningfully to the discourse on sexual consent, it would benefit from addressing legal and cultural dimensions more explicitly, which were noted as absent. However, the article’s findings are relevant and impactful, making it a valuable addition to the literature. With minor revisions, particularly around the study's limitations, the article is worth publishing.

in the Introduction is marked with yellow color

Comments on the Quality of English Language

Overall Evaluation:

The English is generally understandable, but some sentences are overly complex and could be simplified to enhance clarity and flow. There are a few minor issues with subject-verb agreement and awkward phrasing. Addressing these would improve the text's grammatical accuracy. The tone is mostly appropriate for an academic article, but certain parts can benefit from increased precision and conciseness. The ideas are well-structured, but occasionally, the sentence structure could be more direct and less redundant.

With some editing, particularly focusing on conciseness and clarity, the English in the article would be of a high standard.

Objective and Methodology (Lines 306-312):

  • The sentence "The objective of this research was to explore the conceptions of sexual consent among Colombian nationals aged at least 18 years through a qualitative methodology." could be shortened for simplicity:
    Suggestion: "This study aimed to explore Colombian nationals' conceptions of sexual consent, using a qualitative methodology with participants aged 18 and above."
  • themes were identified for each question, resulting in six total themes."

Response: the suggestions were accepted and the changes were made to the text of the article.

Conclusion (Lines 409-419):

  • The sentence "These factors highlight the complexity of sexual consent and its interconnection with various psychological, social, and cultural variables"
  • should be considered in various contexts, including..."

Response: the suggestions were accepted and the changes were made to the text of the article.

REVIEWER 2

Enjoyed reading this well written article. In the discussion - the highlight of how respect is even more fundamental then consent is a great point 

Comments on the Quality of English Language

The minor grammatical/spelling errors are listed below:

line 41 - dYnamic

Response: the suggestions were accepted and the changes were made to the text of the article.

Response: a paragraph was added in the section on instruments where the validation by judges and the administration to 5 people to test the understanding of the questions is commented.

Response: the suggestions were accepted and the changes were made to the text of the article.
